# Breath Ammonia Is a Useful Biomarker Predicting Kidney Function in Chronic Kidney Disease Patients

**DOI:** 10.3390/biomedicines8110468

**Published:** 2020-10-31

**Authors:** Ming-Jen Chan, Yi-Jung Li, Chao-Ching Wu, Yu-Chen Lee, Hsiao-Wen Zan, Hsin-Fei Meng, Meng-Hsuan Hsieh, Chao-Sung Lai, Ya-Chung Tian

**Affiliations:** 1Department of Medicine, Chang Gung University, Taoyuan 333, Taiwan; b9202066@cgmh.org.tw (M.-J.C.); r5259@cgmh.org.tw (Y.-J.L.); 2Kidney Research Center and Department of Nephrology, Linkou Chang Gung Memorial Hospital, Taoyuan 333, Taiwan; menghsieh@cgmh.org.tw (M.-H.H.); cslai@mail.cgu.edu.tw (C.-S.L.); 3Department of Photonics, National Chiao Tung University, Hsinchu 300, Taiwan; yuyet1204@gmail.com (C.-C.W.); 410325041@gms.ndhu.edu.tw (Y.-C.L.); hsiaowen@mail.nctu.edu.tw (H.-W.Z.); 4Institute of Physics, National Chiao Tung University, Hsinchu 300, Taiwan; meng@mail.nctu.edu.tw; 5Department of Electronic Engineering, BioMedical Research Center and Green Technology Research Center, Chang Gung University, Taoyuan 333, Taiwan; 6Department of Materials Engineering, Ming Chi University of Technology, New Taipei City 243, Taiwan

**Keywords:** chronic kidney disease, exhaled ammonia, vertical-channel organic semiconductor sensor

## Abstract

Chronic kidney disease (CKD) is a public health problem and its prevalence has increased worldwide; patients are commonly unaware of the condition. The present study aimed to investigate whether exhaled breath ammonia via vertical-channel organic semiconductor (V-OSC) sensor measurement could be used for rapid CKD screening. We enrolled 121 CKD stage 1–5 patients, including 19 stage 1 patients, 26 stage 2 patients, 38 stage 3 patients, 21 stage 4 patients, and 17 stage 5 patients, from July 2019 to January 2020. Demographic and laboratory data were recorded. The exhaled ammonia was collected and rapidly measured by the V-OSC sensor to correlate with kidney function. Results showed no significant difference in age, sex, body weight, hemoglobin, albumin level, and comorbidities in different CKD stage patients. Correlation analysis demonstrated a good correlation between breath ammonia and blood urea nitrogen levels, serum creatinine levels, and estimated glomerular filtration rate (eGFR). Breath ammonia concentration was significantly elevated with increased CKD stage compared with the previous stage (CKD stage 1/2/3/4/5: 636 ± 94; 1020 ± 120; 1943 ± 326; 4421 ± 1042; 12781 ± 1807 ppb, *p* < 0.05). The receiver operating characteristic curve analysis showed an area under the curve (AUC) of 0.835 (*p* < 0.0001) for distinguishing CKD stage 1 from other CKD stages at 974 ppb (sensitivity, 69%; specificity, 95%). The AUC was 0.831 (*p* < 0.0001) for distinguishing between patients with/without eGFR < 60 mL/min/1.73 m^2^ (cutoff 1187 ppb: sensitivity, 71%; specificity, 78%). At 886 ppb, the sensitivity increased to 80% but the specificity decreased to 69%. This value is suitable for kidney function screening. Breath ammonia detection with V-OSC is a real time, inexpensive, and easy to administer measurement device for screening CKD with reliable diagnostic accuracy.

## 1. Introduction

Chronic kidney disease (CKD) has become an emerging public health problem and a major economic and social burden. The prevalence is estimated to be as high as 11%–13% in the general population, and it is associated with increased cardiovascular diseases and mortality compared to non-CKD population [1,2,3]. Early identification of impaired kidney function and immediate intervention are crucial to delay CKD progression to end-stage kidney disease. However, CKD may be asymptomatic at the early stages and remains mostly undiagnosed. CKD unawareness is, thus, common, even in the developed countries [4]. Nicholas et al. reported that only 29.3% of patients with stage 1 and 22.0% of patients with stage 3 were aware of their CKD [5]. A rapid and low-cost detection method that is suitable for large-scale CKD screening can reduce undiagnosed CKD. Therefore, identifying a non-invasive, easy-to-use, real-time, cost-effective indicator of CKD is crucial for early and rapid detection and prevention of CKD.

The glomerular filtration rate (GFR) requires collection of 24-h urine creatinine amount, and quantification of serum creatinine remains the golden standard to measure kidney function, but this method is inconvenient and time-consuming. Laboratory measurement of serum creatinine with conversion to the estimated glomerular filtration rate (eGFR) and blood urea nitrogen (BUN) levels are commonly used to monitor kidney function, but this requires a blood sample and it is not real-time monitoring. As a novel coronavirus, SARS-CoV-2, causes a highly contagious infection and serious acute respiration disease, which requires quarantine of the public, a biomarker that well correlates with eGFR and can be easily used for CKD screening and monitoring outside the hospital would be useful.

Several exhaled components have been investigated to replace blood tests for rapid monitoring of kidney function [6,7]. One of fast and simple detection methods is exhaled breath ammonia measurement that has been particularly intensively investigated in CKD and dialysis subjects. Ammonia is generated through conversion of nitrogenous substances by bacteria in the gut and the nitrogen cycle in the liver and it is elevated because of reduced excretion in subjects with impaired kidney function. Recent reports proposed that breath ammonia is not related to the serum ammonia, but it is mostly generated by the hydrolysis reaction of salivary urea in the mouth [8,9]. The relationship between breath ammonia and BUN was initially identified. Several research groups focusing on hemodialysis patients reported a high correlation coefficient (i.e., >0.8) between breath ammonia concentration and the BUN level when comparing data before and after dialysis [10,11,12,13].

The breath ammonia detection can be detected by several kinds of methods [9,14,15,16,17,18,19,20,21]. Conventional techniques use mass spectrometry combined with different gas ionization technique such as ion mobility spectrometry (IMS) and selected ion flow tube mass spectrometry (SIFT–MS) [10,12,14,19,20]. IMS measurement uses a weak plasma to ionize gas molecules, and these ionized molecules with different shape, mass, and charge periodically pass a drift tube to form a time-domain spectrum and are specifically identified. SIFT–MS uses the chemical method to ionized molecules, then combined with a mass spectrometry to identify the ionized species. Laser absorption spectrometry is also useful in detecting ammonia with high sensitivity when together with external optical resonator [8,22]. The commonly used cavity ring-down spectrometry (CRDS) employs a high-resolution laser absorption technique together with a unique measurement cell to quantify thousands-of-times increase of the optical path to achieve high sensitivity [20]. All above methods, however, require expensive and bulky devices. Even the laser system is cheaper than the mass spectrometry, the cost is still too high for point-of-care applications. In this study, for the purpose of portable and real-time applications, we used the low-cost disposable solid-state gas sensor. In recent years, there were several studies reported solid-state sensitive ammonia sensors. Some studies successfully showed the result of breath ammonia detection result by using MoO3 nanosensor at 500 °C, CuBr chemiresistor, or reduced graphene oxide with conducting polymer chemiresistor [23,24,25]. Particularly, the polyaniline nanoparticle chemiresistors developed by J. Killard’s team and the vertical-channel organic semiconductor diode (V-OSC sensor) developed by our team were successfully used in clinical trials for CKD patients, indicating the well control of sensor stability and reproducibility [21,26,27]. Here, in this study, the V-OSC sensors with donor-acceptor polymer such as Poly[(4,8-bis(2-ethylhexyloxy)-benzo(1,2-b:4,5-b′)dithiophene)-2,6-diyl-alt-(4-(2-ethylhexyl)-3fluorothieno [3,4-b]thiophene-)-2-carboxylate-2,6-diyl)] (PTB7) were used to realize the breath ammonia detection [26]. The sensing system is portable and the detection requires only 30 s. To suppress the influence of the humidity on breath ammonia detection, the collected exhaled gas was quickly dehumidified within a few seconds and the relative humidity in the sensing chamber was fixed at around 10%. The details of gas collection and system operation are given in the experimental section. The fabrication cost for the sensor chips is extremely low to be able to serve as disposable chips. Previous breath ammonia clinical tests validated the function of the V-OSC sensors. Compared with other methods, this V-OSC sensors have some merits including small size of the device, low cost, and possible point-of-care feasibility.

Although good correlations between breath ammonia and BUN have been demonstrated in hemodialysis patients [11,12,20,26], studies using breath ammonia to predict the occurrence of CKD are rare. One study compared breath ammonia in 27 CKD stage 3–5 patients with 15 healthy subjects [22]. Another study analyzed the difference between breath ammonia in eight CKD stage 4–5 patients and six healthy volunteers [27]. Both studies showed that breath ammonia concentration was significantly higher in CKD patients compared to healthy individuals. Our recent study also demonstrated that breath ammonia concentration was strongly correlated with the BUN level in 34 CKD stage 3–5 patients, and it was a good predictor to distinguish healthy individuals and these CKD patients [28]. However, these studies used healthy subjects as controls and had a small-number size, and their results require verification by large-scale studies. In this study, we determined whether breath ammonia was a useful screening tool to distinguish between all stages of CKD patients in a study with a larger sample size.

## 2. Experimental Section

### 2.1. Patients and Definitions

We enrolled different CKD stages patients in our outpatient clinic of nephrology department from July 2019 to January 2020. One hundred and twenty-one CKD patients including 19 stage 1 patients, 26 stage 2 patients, 38 stage 3 patients, 21 stage 4 patients, and 17 stage 5 patients were enrolled in this study. We excluded patients less than 20 years old. We also excluded the patients with cirrhosis or liver dysfunction, with active gastrointestinal bleeding, with active infection, and with cancer under active treatment. Patients with admission to hospital within one month were also excluded. The CKD categories and the severity of reduced GFR were defined according to the CKD classification of Kidney Disease Outcome and Quality Initiative (K/DOQI) as stage 1 (eGFR ≥ 90 mL/min/1.73 m^2^ and kidney damages), stage 2 (60–89 mL/min/1.73 m^2^; mild GFR decline), stage 3 (30–59 mL/min/1.73 m^2^; moderate GFR decline), stage 4 (15–29 mL/min/1.73 m^2^; severe GFR decline) and stage 5 (<15 mL/min/1.73 m^2^). The demographic characteristics and comorbidities including diabetes, hypertension, congestive heart failure (CHF) and coronary artery disease (CAD) were recorded. Results of laboratory tests including hemoglobin, albumin, BUN, and serum creatinine were collected for analysis. The eGFR was calculated using the Modification of Diet in Renal Disease Study equation. Administered medicine including steroids, angiotensin converting enzyme inhibitor/angiotensin II receptor blocker (ACEi/ARB) and proton pump inhibitor (PPI) were also recorded. All patients provided a written informed consent to participate in this study. Ethics approval was obtained from the Medial Ethics Committee of Linkou Chang Gung Memorial Hospital on 21 June 2018. (IRB number 201800857B0).

### 2.2. Breath Ammonia Collection

The measurement of exhaled ammonia was conducted in the morning. For the sake of convenience in kidney function screening, fasting was not mandatory in order to fit the simplicity of real-world application. To collect the exhaled breath samples, participants were asked to profoundly inhale through the nose and then exhale into a 500 mL plastic bag (the first bag) using a short straw through their mouth. The breath sample in the first bag was soon transferred into a second bag by passing through a desiccation cylinder (that was filled with sodium hydroxide [NaOH]. particles) within 3–5 s. Then, the second plastic bag was immediately connected to the inlet of the Gas Measurement System. It is noted that, when pressing the breath sample through the NaOH tube, NaOH particles also trap a small amount of ammonia. This might cause ammonia concentration deviation in low concentration regime. (i.e., 100–200 ppb)

### 2.3. Gas Measurement System

The gas measurement system used in our previous reports was used in this study [26,27,29]. As shown in Figure 1, the system is composed of a rotameter, pump, desiccation cylinder, the sensing chamber, and an electrical signal measurement instrument (Keysight U2722A USB Modular Source Measure Unit, Keysight Technologies, Santa Rosa, CA, USA). The current of the gas sensor chip (biased at 5 V) inside the sensing chamber was measured in real time using U2722A. The gas sensor chip (Taiwan Patent, 108143915) was reported in our previous research [27,29]. The gas flow rate was fixed at 500 mL/min and ambient air was mixed with the background gas. With continuous pumping of ambient gas to pass through the desiccation cylinder (filled with NaOH), the relative humidity (RH) in the sensing chamber was fixed at approximately 10% and was monitored in real time using a hygrometer. It is noted that the NaOH tube simultaneously suppress the relative humidity and also the CO2 level in the collected sample to avoid the interference due to the formation of carbonic acid in the condensed water film on the sensing layer. When the second bag was connected to the inlet of the system, the breath sample was pumped into the sensing chamber with a fixed RH of approximately 10%. After a fixed sensing time of 30 s, the second bag was removed from the inlet and the background ambient gas entered the chamber again.

### 2.4. Statistical Analysis

Continuous variables are presented as the mean ± standard errors and categorical variables are expressed as the number (%). For comparisons of breath ammonia concentrations between two CKD stages, variables were analyzed using the Student’s *t*-test. For comparisons among five CKD groups, continuous variables were analyzed using a one-way analysis of variance (ANOVA), while categorical variables were compared using the chi-square test (Table 1). *p*-values less than 0.05 were considered to be statistically significant. The accuracy of breath ammonia and BUN to distinguish between the two different categories was evaluated using the receiver operating characteristic (ROC) curve analysis. The correlation analysis used Spearman’s correlation coefficient. Values of *p* < 0.05 were considered to be statistically significant.

## 3. Results

### 3.1. Patient Characteristics

One hundred twenty-one patients with different CKD stages were divided into five groups according to their CKD stages (1–5). Among these patients, 19 patients (15.7%) were CKD stage 1, 26 (21.5%) were stage 2, 38 (31.4%) were stage 3, 21 (17.4%) were stage 4, and 17 (14.0%) were stage 5. The mean eGFR was 116 ± 5.7, 75 ± 1.4, 48 ± 1.3, 20 ± 1.0 and 9 ± 0.6 mL/min/1.73 m^2^, respectively. Demographics and laboratory measurements are summarized in Table 1. The average age of CKD patients was 61.9 ± 15.6 years and patients with CKD stage 1 or 2 were younger than patients with CKD stage 3/4/5 (*p* < 0.001). There was no difference in sex or body weight in the different CKD stage groups. Comorbidities including CHF, CAD, and diabetes did not differ between the groups. The proportion of hypertension, which is a common complication in CKD patients, was significantly higher in the CKD stage 2–5 group compared with the CKD stage 1 group (*p* = 0.010). Hemoglobin and albumin levels were not different in these CKD groups. As expected, BUN levels that were increased in all CKD patients were proportionally increased with CKD progression. The serum albumin level was not different between the groups.

Steroids that increase protein catabolism and subsequently facilitate BUN production were used only in a few patients and the difference in steroid use was similar between the groups. The use of PPIs that suppress gastric acid secretion and reduce exhaled ammonia was not different in the groups.

ACEi and ARB can protect against CKD progression but may temporarily reduce kidney function and cause hyperkalemia, especially in patients with advanced CKD [30,31]. The use of ACEi/ARB in the CKD stage 5 group was significantly lower than that in other CKD groups. To determine whether the use of ACEi/ARB affected breath ammonia, patients were divided into two groups according to the use of ACEi/ARB. The result showed that the breath ammonia concentration was 3146 ± 567 parts per billion (ppb) in the ACEi/ARB group and 3895 ± 815 ppb in the non-ACEi/ARB gourp, which was not significantly different (*p* = 0.442).

### 3.2. Breath Ammonia Concentrations Are Correlated with Kidney Function

Our previous study demonstrated a good correlation between breath ammonia concentration and BUN levels in healthy individuals and CKD stage 3–5 patients [29]. In this study, we assessed the correlation between breath ammonia concentration and BUN levels in all-stage CKD patients including CKD stage 1–2 patients. The result of the Spearman’s correlation analysis demonstrated a strong correlation between breath ammonia concentrations and BUN levels (*R* = 0.723, *p* < 0.0001) (Figure 2A). Similarly, the natural logarithm transformation of breath ammonia was strongly correlated with BUN levels (*R* = 0.756, *p* < 0.0001; Figure 2B).

We further determined the correlation between breath ammonia and kidney function. Breath ammonia concentrations were strongly correlated with serum creatinine levels (*R* = 0.735, *p* < 0.0001; Figure 2C) and negatively correlated with eGFR (*R* = −0.535, *p* < 0.0001; Figure 2D). These results suggest that breath ammonia has a strong correlation with kidney function in all CKD patients regardless of CKD stages.

Our previous study showed a positive correlation between breath ammonia concentration and salivary pH value [29]. The result of the Spearman’s correlation analysis also showed that salivary pH values were significantly correlated with breath ammonia concentrations (*R* = 0.656, *p* < 0.0001; Figure 2E). Salivary pH values were also positively correlated with the BUN levels (*R* = 0.640, *p* < 0.0001; Appendix A) and serum creatinine levels (*R* = 0.677, *p* < 0.0001; Appendix A), but negatively correlated with eGFR (*R* = −0.682, *p* < 0.0001; Appendix A).

### 3.3. Breath Ammonia Concentration Is Increased with CKD Progression

Because progressive decline in kidney function causes accumulation of nitrogen metabolites including ammonia, we determined whether breath ammonia was increased with CKD progression. We first determined breath ammonia concentrations in patients with different CKD stages. Our results showed that the mean concentration of breath ammonia in this study was 3898 ± 6760 and it increased with increasing CKD stages as followed: 636 ± 94 ppb in CKD stage 1 patients, 1020 ± 120 ppb in CKD stage 2 patients, 1943 ± 326 ppb in CKD stage 3 patients, 4421 ± 1042 ppb in CKD stage 4 patients, and 12781 ± 1807 ppb in CKD stage 5 patients (Table 1). The difference in the breath ammonia concentration between each stage was statistically significant and the concentration was increased with increasing CKD stages (Figure 3). This result suggests that breath ammonia concentration could be used to predict CKD stages.

Our previous study demonstrated a simultaneous increase of breath ammonia and salivary pH in advanced CKD patients [29]. To assess whether salivary pH was increased in CKD patients, salivary pH value was measured in our CKD stage 1–5 patients, respectively. As shown in Table 1, salivary pH levels were proportionally elevated with increased CKD stages (*p* < 0.001). The pH value in each stage was significantly higher than that in previous stage and Appendix A).

To determine the predictive value of breath ammonia concentration to distinguish between CKD stage 1 and other CKD stages (2–5), the ROC curve analysis was used. The result showed that the AUC of breath ammonia was 0.835 (*p* < 0.0001). The cutoff value of 974 ppb had a best predictive value to distinguish between CKD stage 1 and other CKD stages, with a sensitivity of 69%, a specificity of 95%, a positive predictive value (PPV) of 0.99 and a negative predictive value (NPV) of 0.36 (Figure 4A). The ROC curve analysis results also demonstrated that the AUC for differentiating CKD stage 1 from early CKD (stage 2 and 3) was 0.751 (*p* < 0.001). At the cutoff value of 974 ppb, the sensitivity was 53%, specificity was 95%, PPV was 0.97, and NPV was 0.38 (Figure 4B).

Because the K/DOQI CKD classification defines CKD as GFR<60 mL/min/1.73 m^2^ for more than three months, the cutoff value of the ROC curve analysis was calculated to distinguish patients between eGFR ≥ 60 mL/min/1.73 m^2^ and patients with eGFR < 60 mL/min/1.73 m^2^. As shown in Figure 4C, the AUC was 0.831 (*p* < 0.0001) with a sensitivity of 71%, a specificity of 78%, a PPV of 0.84, and a NPV of 0.61 at the cutoff value of 1187 ppb.

### 3.4. The Use of Breath Ammonia as a Screening Tool to Predict Patients with CKD

Our findings showed that breath ammonia concentration was increased with the progression of CKD stages and that there was a strong correlation between breath ammonia and kidney function. Thus, we evaluated whether measurement of breath ammonia could be used as a screening tool for the occurrence of CKD. According to the definition of CKD that was established by KDIGO, the patients were divided into two groups by eGFR≥ or <60 mL/min. The sensitivity and specificity of breath ammonia concentration in predicting patients with eGFR < 60 mL/min are shown in Table 2. At a concentration of 993 ppb, the breath ammonia test had a sensitivity of 75% and a specificity of 73%. To increase the test sensitivity to 80%, breath ammonia was set at a concentration of 886 ppb, and the specificity decreased to 69%. This result suggests that for a non-life threating or non-serious CKD, breath ammonia at a cutoff concentration of 886 ppb is a good screening tool for detection of patients with potential CKD.

## 4. Discussion

CKD has an increasing prevalence and incidence, and it is now a worldwide public health problem and a major economic burden. Because early diagnosis is crucial for prevention and delay of adverse CKD outcomes, finding a rapid, low-cost, and non-invasive screening method to identify CKD patients is important. In this study, we used a portable, real-time biosensor to measure exhaled breath ammonia in CKD patients to determine whether breath ammonia could be used as a simple and real-time indicator of kidney function. Our previous study and Wang’s study showed a good correlation between breath ammonia concentration and BUN level in CKD stage 3–5 patients [29,32]. This study enrolled more patients, including CKD stage 1–2 patients, and confirmed the good correlation between breath ammonia and BUN. In addition, breath ammonia concentration was strongly correlated with the serum creatinine level and eGFR. Bayrakli et al. also reported a good correlation between breath ammonia and eGFR [33]. These results all suggest that breath ammonia could be a useful surrogate of kidney function. Measurement of serum creatinine with calculation of eGFR is the most commonly and easily used method to assess kidney function, but it requires blood withdrawal and laboratorial analysis. The real-time detection of the breath ammonia by V-OSC sensor like finger-stinged blood sugar monitoring provides a fast and easy-to-use assessment of kidney function.

Our result demonstrated that breath ammonia concentration was proportionally increased with CKD progression because the concentration at each stage was significantly higher than that at the previous stage (CKD stage 1, 636 ± 94 ppb; stage 2, 1020 ± 120 ppb; stage 3, 1943 ± 326 ppb; stage 4, 4421 ± 1042 ppb; stage 5, 12,781 ± 1807 ppb). Although breath ammonia measurement has been reported to be well correlated with the BUN level in hemodialysis patients, studies that have assessed its use in CKD screening is rare. As per our knowledge, only two studies were performed and they used different methods including an external cavity laser (ECL)-based off-axis cavity-enhanced absorption spectroscopy and an electrochemical gas sensor analyzed breath ammonia concentrations in 27 CKD stage 3–5 and eight CKD stage 4–5 patients, respectively [33,34]. Both studies showed that compared with healthy individuals, these CKD patients had significantly higher breath ammonia levels Our recent study also showed the usefulness of breath ammonia to predict CKD stage 3–5 occurrence. This study enrolled more advanced CKD patients in a different hospital, further demonstrating a significantly increase in the mean breath ammonia concentration with increasing CKD stages [29]. In addition, compared with other studies, the advantage of our device was a portable design and rapid detection (within 5 min) [15,35,36].

The K/DOQI working group defines decreased GFR as GFR < 90 mL/min/1.73 m^2^. Compared with patients with eGFR ≥ 90 mL/min/1.73 m^2^ (CKD stage 1), patients with eGFR < 90 mL/min/1.73 m^2^ (CKD stage 2–5) had higher breath ammonia concentration. The ROC curve analysis showed that breath ammonia is a good predictor (AUC, 0.835) to distinguish between individuals with and without eGFR ≥ 90 mL/min/1.73 m^2^. At a cutoff value of 974 ppb, the sensitivity and the specificity were 69% and 95%, respectively, and there was a high PPV of 0.99, but a low NPV of 0.36. Similarly, breath ammonia is an acceptable indicator (AUC, 0.751) to differentiate between patients with eGFR ≥ 90 mL/min/1.73 m^2^ (CKD stage 1) and patients with a mild to moderate GFR decline (CKD stage 2 and 3) because there was a high specificity (95%) and high PPV (0.97), but there was a low sensitivity (53%) and low NPV (0.38). These results suggest that an elevated breath ammonia concentration more than 974 ppb is a reliable indicator to predict eGFR < 90 mL/min/1.73 m^2^ because 99% of patients with breath ammonia > 974 ppb had eGFR < 90 mL/min/1.73 m^2^. However, 64% of patients with a breath ammonia concentration below 974 ppb may still have eGFR < 90 mL/min/1.73 m^2^. Thus, breath ammonia monitoring can be used to screen asymptomatic CKD patients, but it still requires an eGFR assessment in the high-risk population.

Normal GFR varies with age, and older individuals may have normal kidney physiological function in the range of GFR 60–89 mL/min/1.73 m^2^ [37,38,39]. Because GFR < 60 mL/min/1.73 m^2^ is considered to reflect abnormal kidney function in all ages, the K/DOQI CKD classification defines CKD as GFR < 60 mL/min/1.73 m^2^ for more than three months. Thus, we assessed whether breath ammonia could be used as a predictor to distinguish between patients with and without eGFR < 60 mL/min/1.73 m^2^. The AUC of the ROC curve analysis was 0.831 and at a cutoff value of 1187 ppb, the PPV was 0.84. These results suggest that a breath ammonia concentration > 1187 ppb has a good diagnostic accuracy to verify patients with eGFR < 60 mL/min/1.73 m^2^. However, only 71% of patients with eGFR < 60 mL/min/1.73 m^2^ were identified at this concentration.

For non-fatal disease screening, a good predictor requires a good diagnostic accuracy with adequate sensitivity. To screen for patients with reduced GFR < 60 mL/min/1.73 m^2^, our results showed that a breath ammonia concentration set at 886 ppb had a sensitivity of 80% but the specificity was reduced to 69%. Because serum creatinine is commonly monitored in routine health examination and consequences of false negative and positive results for CKD detection are modest without immediate hazards, a sensitivity of 80% for CKD screening is acceptable. We believe that breath ammonia measurement at this cutoff value could be a useful tool for CKD screening.

ACEi and ARB can cause hyperkalemia and GFR reduction in some patients and no randomized controlled trials have verified the safety and efficacy in patients with advanced CKD [30]. This may explain why, in this study, the proportion of ACEi/ARB use in the CKD stage V group was less than that in other CKD groups. However, there was no significant difference in the breath ammonia concentrations in the ACEi/ARB and non-ACEi/ARB groups in patients with CKD stage 1–4. Obermeier et al. reported that ACEi/ARB did not affect exhaled breath ammonia [6]. Our results are consistent with their finding, suggesting that ACEi/ARB use has no effect on breath ammonia concentration. Steroids are sometimes used in some CKD patients, especially in patients with glomerulonephritis-mediated CKD, and this can increase the metabolic rate and cause an increase in the BUN level. In this study, steroid use was not different in these CKD patients, suggesting that breath ammonia monitoring is useful in CKD patients taking steroids.

Our study also showed that salivary pH was positively correlated to BUN, and breath ammonia concentrations. Salivary pH values were significantly higher in the patients with more advanced CKD stages than earlier stages, which is compatible with our previous study [29]. Ammonia is basic and the increase in salivary pH value could be attributed to the hydrolysis of increased salivary urea to a higher salivary NH3 concentration by bacterial urease in patients with impaired kidney function [40]. The correlation between pH values and breath ammonia has been reported by Schmidt et al. [11]. The breath ammonia (NH3) is formed as the volatile part of NH4+, which is the product of the hydrolysis reaction of salivary urea. As a result, the salivary pH level will affect the concentration ratio between NH3 and NH4+ through the fundamental chemical reaction following Henderson–Hasselbalch equation. Limitations of this study include older age in CKD stage 3–5 patients compared with CKD stage 1–2 patients. However, this may be caused by the nature of CKD and the age factor did not affect breath ammonia concentration (*R* = 0.105, *p* = 0.253, Appendix A). Second, many reports found that the breath ammonia level was affected by food or water intake [11,13]. Thus, most previous studies analyzed 4–6 h-fasting breath ammonia. In this study, fasting was not required, and the effect of fasting was not analyzed. However, for user convenience, the fasting requirement may place a limitation on the real-world application of this screening method. Regardless of the fasting requirement, our data still demonstrated the usefulness of breath ammonia in CKD screening. Finally, despite a larger case number in this study compared with previous studies, enrollment of more subjects is required to verify the usefulness of this CKD screening method. To achieve this goal, we combined the data from 45 CKD stage 1–2 patients and 76 CKD stage 3–5 patients in this study and 34 CKD stage 3–5 patients from our recent study and reanalyzed the data. The results of the re-analyzed ROC curve analysis still demonstrated that breath ammonia measurement was a good method for screening patients with eGFR < 60 mL/min/1.73 m^2^ because the AUC was 0.834, sensitivity was 83%, specificity was 56%, PPV was 0.82, and NPV was 0.57 at a cutoff value of 973 ppb (Appendix A).

## 5. Conclusions

Screening for CKD is important because it is a significant public health burden with a high prevalence worldwide and early diagnosis can delay and reduce morbidity and mortality. Detection of exhaled breath ammonia that is real-time, inexpensive, and easy to administer with an acceptable diagnostic accuracy and sensitivity can be used as a good surrogate of kidney function and a reliable tool for CKD screening.

## Figures and Tables

**Figure 1 biomedicines-08-00468-f001:**
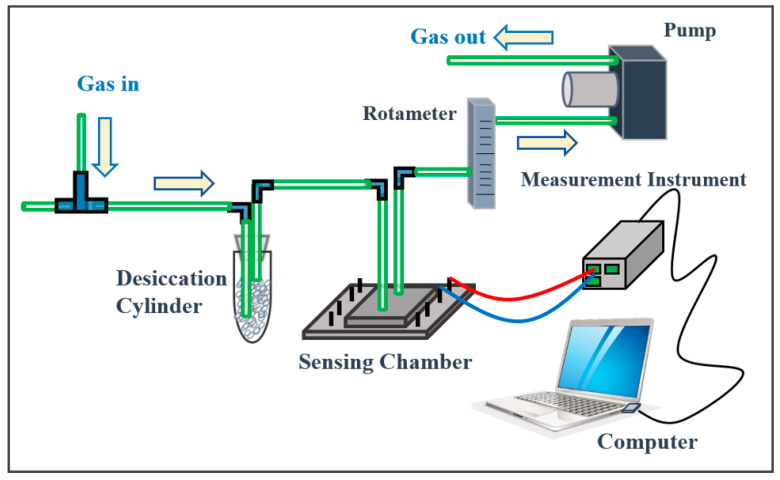
The sensing system includes a desiccation cylinder, an airtight sensing chamber, a rotameter, a pump, and an electrical signal measurement instrument (Keysight U2722A USB Modular Source Measure Unit).

**Figure 2 biomedicines-08-00468-f002:**
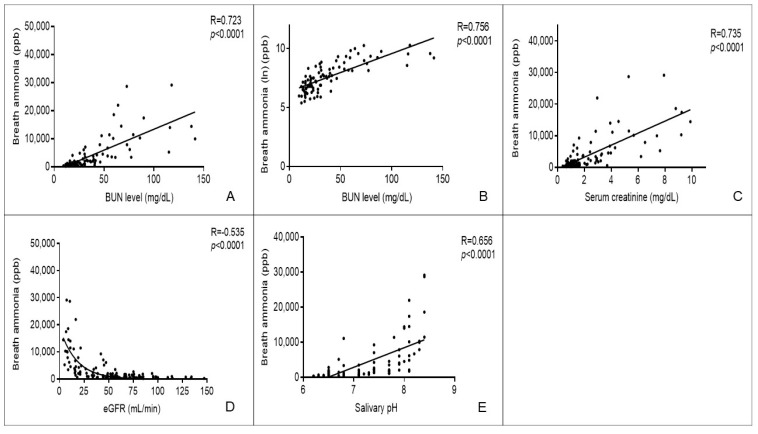
Correlation between breath ammonia and salivary pH value, blood urea nitrogen (BUN), serum creatinine, and estimated glomerular filtration rate. (**A**) Correlation between breath ammonia concentration and BUN level. (**B**) Correlation between breath ammonia concentration (natural logarithm) and BUN level. (**C**) Correlation between breath ammonia concentration and serum creatinine level. (**D**) Correlation between breath ammonia concentration and estimated glomerular filtration rate (eGFR). (**E**) Correlation between breath ammonia concentration and salivary pH.

**Figure 3 biomedicines-08-00468-f003:**
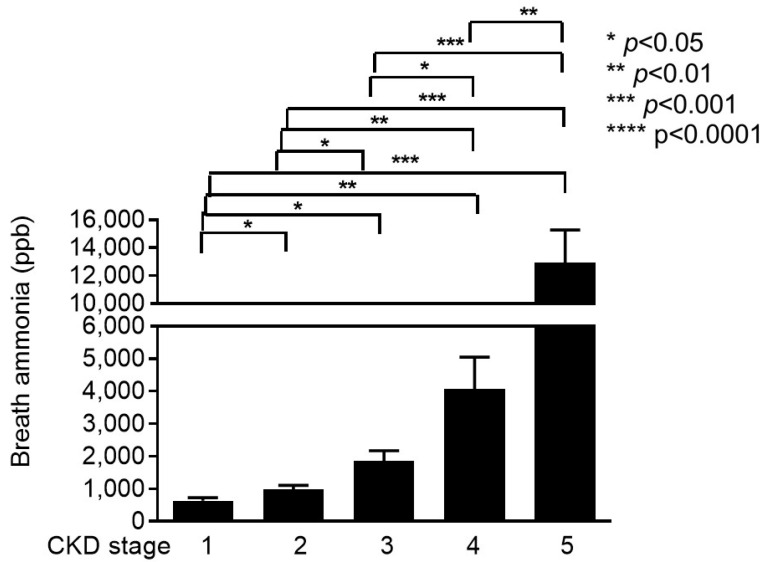
Breath ammonia concentration in patients with chronic kidney disease stage 1–5. Exhaled breath ammonia concentrations were measured in chronic kidney disease (CKD) stage 1–5 patients. Box plot is used to display breath ammonia concentration in different CKD group, and significant differences between groups are indicated (*, *p* < 0.05; **, *p* < 0.01; ***, *p* < 0.001; ****, *p* < 0.0001).

**Figure 4 biomedicines-08-00468-f004:**
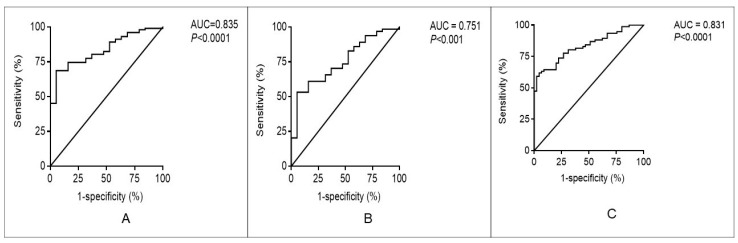
The receiver operating characteristic curve analysis to determine the predictive value of breath ammonia concentration to distinguish between different chronic kidney disease groups. (**A**) The receiver operating characteristic curve analysis for breath ammonia to distinguish between CKD stage 1 and other CKD stages (2–5). (**B**) The ROC curve analysis for breath ammonia to distinguish between CKD stage 1 patients and CKD stage 2 and 3 patients. (**C**) The ROC curve analysis for breath ammonia to distinguish patients between patients with eGFR 60 mL/min/1.73 m^2^ and patients with eGFR < 60 mL/min/1.73 m^2^. AUC: area under curve.

**Table 1 biomedicines-08-00468-t001:** Patient characteristics by different chronic kidney disease (CKD) stages.

Variable	Stage 1 (*n* = 19)	Stage 2 (*n* = 26)	Stage 3 (*n* = 38)	Stage 4 (*n* = 21)	Stage 5 (*n* = 17)	All (*n* = 121)	*p*-Value
Age (year)	49.4 ± 4.1	58.3 ± 2.3	66.4 ± 2.3	66.4 ± 3.2	66.0 ± 3.6	61.9 ± 1.4	<0.001
Male (%)	8 (42.1%)	16 (61.5%)	19 (50%)	11 (52.4%)	8 (47.1%)	62 (51.2%)	0.756
Body weight (Kg)	68.2 ± 3.4	70.1 ± 3.0	63.2 ± 1.7	64.3 ± 2.6	60.2 ± 2.0	65.3 ± 1.1	0.067
Hemoglobin (g/mL)	12.8 ± 0.7	14.9 ± 0.4	15.4 ± 1.6	13.9 ± 1.6	9.8 ± 0.3	13.8 ± 0.7	0.089
Albumin (g/dL)	4.29 ± 0.16	4.19 ± 0.11	4.19 ± 0.08	4.04 ± 0.09	3.99 ± 0.09	4.12 ± 0.04	0.251
BUN (mg/dL)	15.6 ± 1.9	17.7 ± 1.1	24.4 ± 1.6	42.6 ± 2.5	85.1 ± 7.5	38.7 ± 3.1	<0.001
Creatinine (mg/dL)	0.65 ± 0.03	0.95 ± 0.04	1.34 ± 0.04	2.87 ± 0.12	6.46 ± 0.50	2.13 ± 0.19	<0.001
eGFR (mL/min)	116 ± 5.7	75 ± 1.4	48 ± 1.3	20 ± 1.0	9 ± 0.6	54 ± 3.3	<0.001
Breath ammonia (ppb)	636 ± 94	1020 ± 120	1943 ± 326	4421 ± 1042	12781 ± 1807	3493 ± 484	<0.001
Salivary pH	6.52 ± 2.68	6.84 ± 0.50	7.13 ± 0.54	7.61 ± 0.48	8.10 ± 0.41	7.21 ± 0.68	<0.001
Comorbidity							
CHF	0 (0%)	1 (3.8%)	1 (2.6%)	1 (4.8%)	0 (0%)	3 (2.5%)	0.814
CAD	1 (5.3%)	2 (7.7%)	4 (10.5%)	0 (0%)	4 (23.5%)	11 (9.1%)	0.142
HTN	8 (42.1%)	21 (80.8%)	28 (73.7%)	19 (90.5%)	12 (70.6%)	8 (72.7%)	0.010
DM	7 (36.8%)	11 (40.7%)	13 (34.2%)	9 (42.9%)	6 (35.3%)	46 (38.0%)	0.950
Medication							
Steroids	6 (31.6%)	3 (11.5%)	5 (13.2%)	3 (14.3%)	2 (11.8%)	19 (15.7%)	0.358
ACEi/ARB	8 (42.1%)	20 (76.9%)	21 (55.3%)	10 (47.6%)	6 (35.3%)	65 (53.7%)	0.053
PPI use	0 (0%)	0 (0%)	2 (5.3%)	0 (0%)	1 (5.9%)	3 (2.5%)	0.447

CKD, chronic kidney disease; BUN, blood urea nitrogen; CHF, congestive heart failure; CAD, coronary heart disease; DM, diabetes mellitus; eGFR, estimated glomerular filtration rate; ACEi, angiotensin converting enzyme inhibitor; ARB, angiotensin receptor II blocker; PPI, proton pump inhibitor.

**Table 2 biomedicines-08-00468-t002:** Sensitivity and specificity of breath ammonia in predicting patients with <60 mL/min/1.73 m^2^.

Ammonia	Sensitivity	Specificity
680	0.868	0.489
690	0.855	0.489
701	0.842	0.489
710	0.842	0.511
720	0.842	0.533
744	0.829	0.533
765	0.829	0.556
770	0.816	0.556
775	0.816	0.578
782	0.816	0.600
812	0.816	0.622
849	0.803	0.622
870	0.803	0.644
881	0.803	0.667
886	0.803	0.689
888	0.789	0.689
907	0.776	0.689
944	0.776	0.711
974	0.776	0.733
984	0.763	0.733
993	0.750	0.733
1056	0.737	0.733
1134	0.737	0.756
1188	0.737	0.778
1246	0.724	0.778

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
