# Peer review of "Breath Ammonia Is a Useful Biomarker Predicting Kidney Function in Chronic Kidney Disease Patients"

_biomedicines, 2020, doi:10.3390/biomedicines8110468_

Round 1

Reviewer 1 Report

The authors have mostly responded to my criticism in an adequate manner. However, I have two additional major comments which have to be taken into account in the manuscript:

1) I do not understand how the dehumidifying process can fundamentally help in the ammonia adsorption. Pleasee see, for example, Vaittinen, O., Metsälä, M., Halonen, L. et al. Effect of moisture on the adsorption of ammonia. Appl. Phys. B 124. Ammonia adsorption actually increases when humidity decreases (magnitude of change is different for different materials).  Ammonia will adsorb to surfaces even without the presence of water. For this reason, optimal conditions for ammonia analysis are: flow measurement, increased temperature, use of inert materials. Dehumidification can probably help by reducing the changes in water concentration which can induce sudden changes in ammonia adsorption but constant humid conditions are actually best (see ref. above). But maybe the sensor that the authors use does not perform well in humid conditions?

2) It's very nice that the authors have included the pH data and discussion. However, they are not mentioning the main effect to salivary pH which is to shift the acid-base balance in saliva. The higher the pH, the more there is of ammonia in the volatile NH3 form (and less as NH4+). So, if the saliva pH increases, the gas phase ammonia concentration goes up, even if total ammonia concentration (in saliva) stays the same. Please see details in Ref. 11 (Schmidt et al.). Therefore, it can be that much of the NH3 variation between the groups can be actually be explained by salivary pH. This must be discussed in the manuscript.

Author Response

The revised manuscript is in the attached file

Q1) I do not understand how the dehumidifying process can fundamentally help in the ammonia adsorption. Pleasee see, for example, Vaittinen, O., Metsälä, M., Halonen, L. et al. Effect of moisture on the adsorption of ammonia. Appl. Phys. B 124. Ammonia adsorption actually increases when humidity decreases (magnitude of change is different for different materials).  Ammonia will adsorb to surfaces even without the presence of water. For this reason, optimal conditions for ammonia analysis are: flow measurement, increased temperature, use of inert materials. Dehumidification can probably help by reducing the changes in water concentration which can induce sudden changes in ammonia adsorption but constant humid conditions are actually best (see ref. above). But maybe the sensor that the authors use does not perform well in humid conditions?

R1

We would like to thank reviewer for the comment. The literature that the reviewer referred to was testing hydrophobic material (treated stainless steel and PTFE etc.) surface under relative dry condition where water concentration was at ppm level. Under given test condition, the limited amount of NH3 and H2O compete for surface adsorption sites that is why they concluded that NH3 can adsorb on surface even without the present of water. In our work, the sensor exhibits a nanoporous structure covering with a very thin (~ 30 nm) organic semiconductor material (PTB7). The nanoporous structure and the polar parts in PTB7 make water molecules being easily absorbed onto the sensing surface. The surface properties of our sensing material are significantly different than stainless steel or Teflon. The exhale gas steam from human body is overly saturated with water (i.e., near 100 % RH with excess H2O aerosol). If the water adsorption is not properly controlled, the condensed water film can induce unpredictable change in electrical properties of the film. Also, in our previous work [2013, Analytical Chemistry, https://doi.org/10.1021/ac303100k], we reported the importance to suppress CO2 level at the same time when reducing the humidity. We noticed that, with high humidity and high CO2 concentration, the formation of carbonic acid in the condensed water film on the organic semiconductor sensing layer caused stability issue in the sensor performance. This is the reason why we chose to use NaOH tube to simultaneously control the relative humidity at around 10% and suppress the CO2 concentration in the collected exhaled gas sample. We have added the short explanation in the Method section (page 4 line 26-28). (It is noted that NaOH tube simultaneously suppress the relative humidity and also the CO2 level in the collected sample to avoid the interference due to the formation of carbonic acid in the condensed water film on the sensing layer.)

Q2) It's very nice that the authors have included the pH data and discussion. However, they are not mentioning the main effect to salivary pH which is to shift the acid-base balance in saliva. The higher the pH, the more there is of ammonia in the volatile NH3 form (and less as NH4+). So, if the saliva pH increases, the gas phase ammonia concentration goes up, even if total ammonia concentration (in saliva) stays the same. Please see details in Ref. 11 (Schmidt et al.). Therefore, it can be that much of the NH3 variation between the groups can be actually be explained by salivary pH. This must be discussed in the manuscript.

R2 We thank you reviewer for the suggestion. We added the discussion of pH level effect on the formation of volatile NH3 in the manuscript text (page 11, line 45-49).

Reviewer 2 Report

  1. Table 1 can easily be described using 1-2 sentences, so it’s not necessary to have a table in the introduction since it’s already long.
  2. 1, in general, we put the enrollment criteria in the beginning, and the exclude criteria afterward.

Author Response

Q1. Table 1 can easily be described using 1-2 sentences, so it’s not necessary to have a table in the introduction since it’s already long.

R1. Thank you for your comment. We have changed the manuscript accordingly and deleted the table 1.

Q2, in general, we put the enrollment criteria in the beginning, and the exclude criteria afterward.

R2. We would like to thank the reviewer for the suggestion. We have changed the sequence of the description on page 3 line 25-27.

This manuscript is a resubmission of an earlier submission. The following is a list of the peer review reports and author responses from that submission.

Round 1

Reviewer 1 Report

The manuscript shows a correlation between exhaled breath ammonia levels and CKD stage. As such, the data is interesting and valuable for future development of ammonia breath tests.

However, I have a few comments that the authors should take into account before the manuscript can be accepted for publication.

Major comments:

  • The authors correctly note that that SIFT-MS is a bulky and expensive method. CRDS is cheaper and can be made smaller but is probably not yet ready for point-of-care. IMS instruments are also much smaller and cheaper than SIFT-MS. In that sense a cheap & small NH3 would be very welcome if the performance would be adequate. However, many of these have be demonstrated, see for example Gouma, Perena, et al. "Nanosensor and breath analyzer for ammonia detection in exhaled human breath." IEEE Sensors Journal 10.1 (2009): 49-53 (and many others). The authors should comment how their sensor compares with other NH3 sensors, pros and cons.
  • Ammonia is a very sticky molecule and quantitative measurement should optimally be performed in real time in a fast gas flow. The procedure to sample the breath into bag and then suck it slowly onto the sensor is far from optimal. There is a lot of surface area present for the ammonia molecules to adsorb (is the whole gas path heated? what are the materials exactly?) It would be very nice if the authors could compare the performance of their setup with a technique that is known to perform well in NH3 analysis, such as CRDS. If this is not possible, the authors should at least acknowledge these shortcomings and comment on them in the manuscript. This is important since the technical shortcomings probably explain partly why the correlations are not better and why the ROC curves are not optimal.
  • I’m disappointed that the authors did not incorporate simultaneous saliva analysis (for example pH) along with the breath measurements. As they correctly note, it has been shown that ammonia originates mainly from the oral cavity (via hydrolysis of urea) and pH of saliva plays a big role here. If they had taken the pH into account, the numbers would probably be much better. This must be addressed in the manuscript.

Minor comments:

  • The title should probably be “…chronic kidney disease patients”
  • Page two, end of line 40, should be “healthy”
  • Page two, middle of line 26, should be "healthy"

Reviewer 2 Report

The authors in Breath ammonia is a useful biomarker predicting kidney function in chronic kidney disease patient have developed a rapid and convenient method and a specific system to detect/differentiate the different stages of CKD.  It will be a good benefit for the early diagnose and monitor the progress of CKD. There are still some following questions that need to be improved.

  1. More details of methods are needed to be added in the Abstract. The number of subjects should be added as well.
  2. The Introduction can be trimmed, for example, it’s not necessary to introduce the traditional methods, instead, the authors can just talk about directly their disadvantages.
  3. The authors need to put all items of the eligibility criteria together, like the definition of different CKD stages should be in right after the enrollment time introduction. Please reorganize the sentences in the paragraph of 2.1.
  4. Please include the number of participants in 2.1.
  5. Page 3, Line 18, what time were the participants asked to do the measurement? Since fasting wasn’t required, the time is important too.
  6. Please add the (P</>0.05) right after the description sentences in the Result text.
  7. Page 8, Line 17, What are the specificity and sensitivity of the traditional methods? Is there any way to compare it with them?
  8. Page 8, line 42, “As per our knowledge” needs to be added before “only two studies…”